# The impact of augmented feedback (and technology) on learning and teaching cricket skill: A systematic review with meta-analysis

**Kevin Tissera**[1]*, **Dominic Orth**[2], **Minh Huynh**[3], **Amanda C. Benson**[1]

**1** Department of Health Sciences and Biostatistics, Sport Innovation Research Group, Swinburne University of Technology, Melbourne, Australia, **2** Department of Health Sciences and Biostatistics, Swinburne University of Technology, Melbourne, Australia, **3** School of Allied Health, Human Services & Sport, La Trobe University, Melbourne, Australia

☯ These authors contributed equally to this work.

* ktissera@swin.edu.au

**Data Availability Statement:** All relevant data are within the manuscript and its Supporting Information files.

## Abstract

Augmented feedback, including that provided using technology, can elicit multifaceted benefits on perceptual-motor learning and performance of sporting skills. However, current considerations of the applied value in supporting learning and teaching cricket skill is limited. This systematic review with meta-analysis aimed to understand the role and effectiveness of feedback-involved interventions on skill-based performance outcomes in cricket-related research. Six electronic databases were searched (SPORTDiscus, CINAHL, MEDLINE, Scopus, Web of Science and PsycINFO). Of 8,262 records identified, 11 studies met inclusion criteria; five of which were included in meta-analyses. Given no studies with an isolated feedback intervention-arm were identified, the two meta-analyses explored anticipation-based studies consisting of an intervention that included augmented feedback; positioned with respect to the key motor skill concepts of perception (anticipation accuracy) and action (performance success). Despite results highlighting improved performance outcomes for the feedback-involved intervention groups, with a large effect size for improved anticipation accuracy (*Hedge's g* = 1.21, 95% CIs [-0.37, 2.78]) and a medium effect size for overall performance success (*Hedge's g* = 0.55, 95% CIs [-0.39, 1.50]), results were not statistically significant and should be interpreted with caution given the wide confidence intervals. Considering the small number of studies available, in addition to the lack of isolated feedback protocols, further research is warranted to thoroughly explore the impact of augmented feedback on skill-based performance in cricket. Beyond the meta-analyses, the review also explored all included studies from an ecological dynamics perspective; presenting future avenues of research framed around evaluating the applied value of using augmented feedback (mediated with or without technology) for learning and teaching skill in cricket.

**Trial registration:** The protocol was preregistered with Open Science Framework (osf.io/384pd).

**Funding:** The lead author was supported by an Australian Government Research Training Program (RTP) Scholarship. No other funding was received.

**Competing interests:** The authors have declared that no competing interests exist.

## Introduction

Cricket is a popular global sport [1] played by males and females at elite and community levels [2]. The game involves specialised skills within the dynamic tasks of batting, bowling, and fielding. To facilitate learning and teaching of skilled behaviours, technology can play an important and positive role. For example, smart device applications that enable quick analysis of movement patterns, such as the bat lift in cricket [3], provide augmented information that can be delivered as feedback (either directly to a player or by a coach). This extrinsic information can subsequently influence how practice can translate into improved performance outcomes during competition [4].

Augmented feedback is information that is modified (by technology and/or a coach) about an action or action outcome. It can be delivered to a learner during or after an action is completed, putatively, to improve performance during later actions [4]. The learning effects of augmented feedback have been previously examined, including the effect of manipulating '*what*' (type of feedback), '*when*' (timing and frequency), and '*how*' (modality) feedback is provided during practice [5]. Contemporary principles of using feedback to improve perceptual-motor learning [6–8] include: *reducing* the frequency of feedback [9]; allowing for a margin of error (or bandwidth) around the to-be-learned movement pattern or the target outcome [10], and; allowing the learner to control when feedback is obtained [11]. In contrast, there are also negative aspects reported of providing feedback on learning. For example, learning (as well as motivation) can be degraded when augmented feedback is given too frequently [12], in ways that emphasise poor performance [13, 14], or in ways that strip the athlete of control over when the feedback is delivered [15].

Applied learning and teaching frameworks (e.g., non-linear pedagogy, constraints-led approach, ecological dynamics) have interpreted these positive aspects of augmented feedback on learning as facilitating processes that characterise expertise acquisition [16]. Specifically, feedback improves learning in how it facilitates exploration toward identifying more useful information to regulate goal-supportive movement [17, 18]. This is particularly critical in sport contexts, such as in cricket, where there is a need for a tight coupling between information and movement, and where significant variation in task, individual, and environmental constraints creates uncertainty around what and when information-movement couplings can be useful to the athlete [19–21]. Augmented feedback is therefore potentially useful in dynamic contexts that require a period of exploration to help the athlete discover and/or attune to performance-relevant information [22, 23].

Negative aspects of augmented feedback (particularly when delivered via technology at 100% schedules) have also been identified. For example, Woods et al. recently applied ecological dynamics to outline how technology use can disengage the user from the performance environment to, "replace and intervene in direct human perception-action interactions with the environment" [24]. An over-reliance on technology (for feedback) would therefore lead to a deskilling in the athlete's capability to manage unpredictability, uncertainty, and variation in the environment.

In cricket, there are currently no clear guidelines (or systematic reviews on the topic) for using feedback in learning and performance contexts. This lack of clear professional practice guidelines is despite the proliferation of technology innovations (such as microsensor-based technologies [25, 26], video-based technologies [3, 27] and, ball-tracking technologies [28]) currently available for use as augmented feedback during athlete training. This is a significant gap considering that translating how to apply and use technology-enabled feedback in ways that promote its positive aspects on learning (and prevent its negative aspects) remains a challenge both in applied contexts as well as the broader sport science literature [29–31].

Therefore, the primary objective of this systematic review with meta-analysis was to examine how augmented feedback is currently used within cricket-related research. Specifically, this systematic review evaluates the effectiveness of interventions utilising augmented feedback, and their impact on improving skill-based performance. A secondary objective for this review was to evaluate current work using the ecological dynamics framework and derive recommendations and future research directions for using augmented feedback, with and without technology, for learning and teaching skill in the sport of cricket.

## Methods

### Protocol and registration

This systematic review was designed with the Preferred Reporting Items for Systematic Reviews and Meta-Analyses (PRISMA) guidelines [32]. The protocol was preregistered with Open Science Framework (osf.io/384pd).

### Design

The search was conducted by one author (KT) using the SPORTDiscus, CINAHL, MEDLINE, Scopus, Web of Science and PsycINFO academic databases, from the earliest available entry until 12 May 2022. Search terms consisted of the activity (cricket) and feedback-related terminology, with exclusions for insect and animal-based studies (for full search strategy, please see S1 Table). Manual searching of the reference lists of identified articles were also undertaken, in addition to a Google Scholar search of lead authors within the field.

### Inclusion and exclusion criteria

Inclusion criteria involved full-text, English peer-reviewed studies. At least one arm of the training intervention had to provide a form of augmented feedback to the participant (cricket player), with a focus on improving an area of skill-based performance. Further, any skill level, age or sex of cricketer was permitted. For this review, augmented feedback is defined as any instruction or information provided to the participant that is external to the information they inherently receive from the task itself (i.e., task-intrinsic information) [33]. The augmented feedback may be provided during performance (i.e., concurrent feedback) or provided at the end of the performance (i.e., terminal feedback). Study designs included randomised controlled trials (RCT's), non-randomised controlled trials (non-RCT's), uncontrolled studies, crossover studies and case studies. Qualitative studies with no measure of performance, systematic reviews, animal research studies, conference abstracts and other non-full text articles or non-research publication (commentaries, letters, guidelines, thesis, textbooks) were excluded.

References were imported to EndNote X9.2 software for data management (Clarivate Analytics, Philadelphia, USA). After duplicate articles were removed, titles and abstracts were screened independently by two reviewers (KT and either AB or DO) using the web-based Rayyan systematic review tool (2016, Qatar Computing Research Institute, Doha, Qatar) [34], based on the inclusion criteria. Full text of potentially eligible studies was retrieved, and assessed independently by two reviewers (KT, AB) based on the exclusion criteria. If any discrepancies arose, the third reviewer made a final decision (DO).

### Data extraction

Data were extracted into an Excel document by one author (KT), with remaining authors checking for accuracy. Information relating to the key characteristics (study settings, population information, study interventions and outcomes, type of feedback) was organised into a

table (Table 1). If data were missing, authors were contacted requesting additional information. Further, data were extrapolated manually from graphs where required, using WebPlotDigitiser software (Version 3.12, Austin, Texas, USA).

## Assessment of methodological quality and risk of bias

Given the experimental and non-randomised nature of the study designs for the majority of the included studies, the Downs and Black checklist [35] was deemed appropriate as an accepted approach to assess quality and risk of bias [36]. The Downs and Black checklist was modified in line with other versions within health science research [37, 38]; with Question 27 adjusted to "Did the study have sufficient power?", with one point awarded for the inclusion of a power calculation. Subsequently, a total of 28 points were available. For quality assessment comparisons, study appraisal scores were graded based on the following Kennelly categories [39], in line with previous studies that utilised the modified Downs and Black checklist [37, 40]: good ($\geq$20), fair (15–19), or poor ($\leq$14). Further, six question's (16, 18, 20, 21, 22 and 25) from the checklist which assessed internal validity (bias and confounding) were used to determine a risk of bias score [37]. A total score greater than or equal to 4/6 (67%) resulted in the classification of low risk of bias. Two authors (KT and DO) independently assessed methodological quality and risk of bias, with any discrepancies addressed through consultation with another author (AB).

## Statistical analysis

Two meta-analyses were conducted to understand how current research has explored the impact of augmented feedback on skill-based performance in cricket. Given the limited studies identified (see Fig 1), and none with an isolated feedback-arm, the studies included in the meta-analyses explored occlusion-based anticipation studies in which augmented feedback was an important element of the training intervention. Further, the analyses focused on perception (anticipation accuracy) and action (performance success) outcomes, key attributes of skill-based performance when viewed through an ecological dynamics theoretical framework. Where appropriate, augmented feedback modalities were combined to enable a comparison of intervention (feedback) and control (no-feedback) groups; similar to a previous meta-analyses exploring augmented feedback [41].

A random-effects model was used for analysis to account for differences in both study populations and intervention protocols [36]. Both meta-analyses were performed using the Meta package in R (v 4.0.2; R Core Team, https://www.r-project.org/). The mean, SD, and total number of samples per study for each meta-analysis was loaded as a data frame in R. Where required, standard deviation was calculated from reported levels of standard error [42–44]. To create pairwise comparisons, intervention group data were combined [42, 44, 45]. For one study [44], where results for ball type (full and short) where split, an initial within group combination was required to enable comparison. Combined ball type mean was calculated (Formula 1, Formula 2) using the recommended formula in the Cochrane guidelines [36]:

$$Combined\ group\ mean = \frac{(N_1 M_1) + (N_2 M_2)}{(N_1 + N_2)} \qquad \text{Formula 1}$$

where N equals the number of participants and M equals the mean for the full or short ball for each group.

$$Combined\ group\ SD = \sqrt{\frac{(SD_1^2 + SD_2^2)}{2}} \qquad \text{Formula 2}$$

where SD equals the standard deviation for the full or short ball for each group.

**Table 1. Characteristics of studies included in systematic review and meta-analysis.**

| Reference (Country) | Participants | Cricket skill | Outcomes | Intervention & Control Groups | Feedback type | Feedback provided |
|---|---|---|---|---|---|---|
| Barker et al. [49]**, United Kingdom | N = 1, age = 21 y, male, sub-elite (semi-professional) | Bowling | 1. Trait self-confidence, 2. Self-efficacy, 3. ARS self-confidence, 4. Bowling average, 5. Bowling strike rate, 6. No. wickets taken | 1. Hypnosis, Technique refinement & Self-Modelling (no control as a single-case study) | Verbal Visual | Video replay of adaptive behaviour (good performance, technique development, consistency, effort) for self-modelling |
| Brenton et al. [42]*, Australia | N = 39, mean = 23.85 y, all male, sub-elite (district cricket clubs) | Batting | 1. Accuracy of Anticipation (%) 2. Transfer test for both a) fast and b) slow bowling types | 1. Visual-perceptual (VP) training, 2. Visuomotor pattern (VM) training, 3. Control group | Visual | VP = Unoccluded replay following occluded trial, VM = Unoccluded replay following occluded trial and observing trajectory of ball after bowling attempt Control = No feedback |
| Brenton et al. [43]*, Australia | N = 12, mean = 22.85 y, all male, elite (state cricket squad) | Batting | 1. Accuracy of Anticipation (%), 2. Transfer test for both a) fast and b) slow bowling types, 3. Match transfer | 1. Visuomotor pattern (VM) training, 2. Control group | Visual | VM = Unoccluded replay following occluded trial and observing trajectory of ball after bowling attempt Control = No feedback |
| Fuss et al. [50]**, Australia | N = 1, age = not included, sex not included, playing standard not included | Bowling | 1. Physical Performance Parameters (spin rate, angular acceleration, resultant torque, spin torque, power), 2. Skill Performance Parameters (precession, normalised precession, precession torque, efficiency, frequency) | 1. Training intervention with feedback (no control as a single-case study) | Verbal Visual | Verbal feedback during the training intervention and visual feedback via modelling based on smart ball data |
| Hopwood et al. [51]*, Australia | N = 12, mean = 21.3 y, all male, elite (Australian Cricket Academy [AIS CoE]) | Fielding | 1. Decision Accuracy (%), 2. Fielding Success (%), 3. Mean Movement Initiation Time (msec) | 1. Perceptual training group (PT), 2. Control group | Visual | PT = Unoccluded video replay following occluded trial, Control = No feedback |
| Middleton et al. [52]**, Australia | N = 1, age = withheld, sex withheld, elite (international) | Bowling | 1. Elbow flexion-extension angle | 1. Bowling remedial work (no control as a single-case study) | Verbal Visual | Video footage to support coaching instructions |
| Muller et al. [44]*, Australia | N = 23, mean = 21.52 y, all male, sub-elite (district cricket clubs) | Batting | 1. Prediction accuracy for ball type (%), 2. Foot-movement accuracy (%), 3. Bat-Ball Contact Accuracy (%) | 1. With-movement [WM], 2. Without-movement [WoM], 3. Control group | Verbal | WM = indication of delivery type and direction the ball was struck following occluded trial, WoM = indication of the delivery type following occluded trial, Control = No feedback |
| Neil et al. [53]**, United Kingdom | N = 1, age = 19 y, male, junior-elite (Youth Academy) | Batting | 1. Self-efficacy rating, 2. Batting average | 1. Guided self-reflection (no control as a single-case study) | Verbal Visual | Video replay of batting performance, verbal prompts for self-guided reflection |
| Ranson et al. [54]***, United Kingdom | N = 14, mean = 18.5 y, all male, elite (England and Wales Elite Fast Bowling Group) | Bowling | 1. Shoulder alignment (°), 2. Back knee (°), 3. Front knee (°), 4. Maximum lower trunk motion (°), 5. Ball velocity | 1. Coached, 2. Not-coached | Verbal Visual | Verbal cues and video footage provided; however, specifics not outlined due to study length (two years) and variety of coaches involved |
| Smeeton et al. [45]*, United Kingdom | N = 33, mean = 14.9 y, all male, junior-elite (county cricket standard) | Batting | 1. Response Accuracy (%), 2. Response Time (msec) | 1. Video Replay (V), 2. Imagery No Replay (I), 3. Outcome KR No Replay (O), 4. Control group | Verbal Visual | V = Player told spin direction, cue instructions, watch delivery replay (no occlusion), I = Player told spin direction, cue instructions, player images delivery, O = Player told spin direction, Control = No feedback |
| Wallis et al. [55], Australia | N = 33, mean = 13 y, all male, junior (cricket specialist schools) | Bowling | 1. Biomechanical outcomes (flexion-extension angles), 2. Bowling accuracy (only for harness group), 3. Ball release velocity | 1. Harness Group (H), 2. Non-harness Group (NH) | Verbal Visual | H = bowling harness, coaching cues, video analysis & mirrors NH = coaching cues, video analysis & mirrors |

*Included in meta-analysis; **Single case study with no control group; ***Intervention group relates to one of four specific areas of coaching (More Side-on; Less Back Knee Flexion; Less Front Knee Flexion; Less Lower Trunk Side-flexion)–participant may have been coached on one attribute, but not another. Feedback provision not specified in this study.

For each outcome, effect sizes were computed using standard mean difference, expressed as *Hedge's g*, given the small sample sizes [46]. The magnitude of the effect sizes (0.2 = small, 0.5 = medium, 0.8 = large) were interpreted in accordance with Cohen [47]. Heterogeneity was measured using the $I^2$ statistic, interpreted according to criteria previously set out [36]: low (0–25%), moderate (26–74%) and high (75–100%). High heterogeneity suggests that there is greater variation in the assumed intervention effects [36]. Influence analysis was used to identify extreme effect sizes (i.e., outliers) and examine if some studies exert high influence in the overall results [48]. A $p < 0.05$ was considered to be statistically significant.

## Results

### Search results

The literature search resulted in 8,262 records from six databases. The PRISMA flow diagram of search strategy, study selection and exclusion reasoning are presented in Fig 1. A total of 11 studies were included in qualitative synthesis [42–45, 49–55]. Two studies incorporated a study design that consisted of multiple experiments [50] or case studies [53]. For these, only the experiment or case study relevant to this systematic review (i.e., featuring augmented feedback) was included and reported on. Five of the identified studies were included in the meta-analyses, given the ability to combine similar occlusion-based anticipation studies which involved augmented feedback as part of the training intervention [42–45, 51]. No studies identified in the search included an isolated feedback intervention arm.

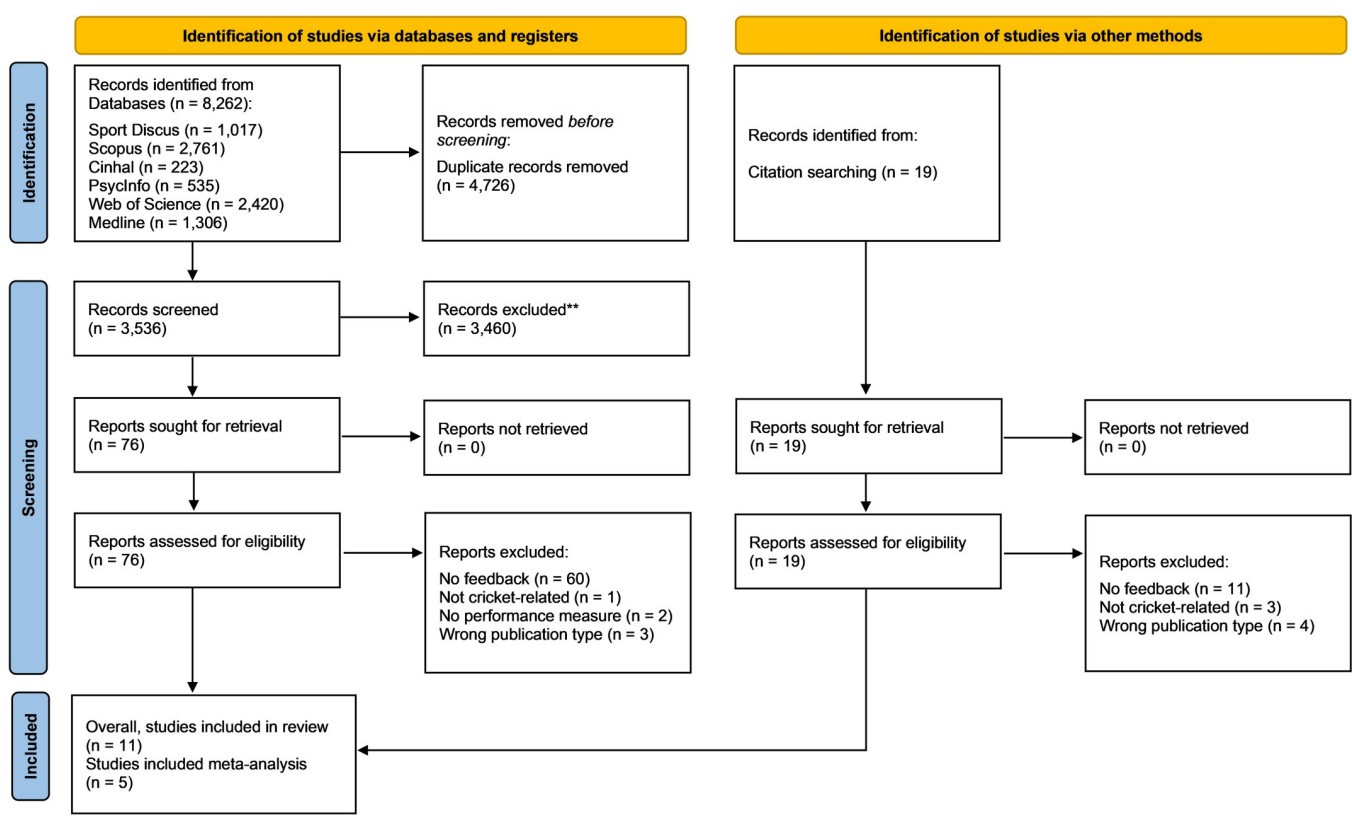

**Fig 1. PRISMA flowchart of search strategy, study selection and exclusion reasoning.**

## Study characteristics

Characteristics of the 11 studies included in the systematic review are summarised in Table 1. Seven studies were conducted in Australia [42–44, 50–52, 55], with the remaining four from the United Kingdom [45, 49, 53, 54]. Two single-case studies did not include participant information (n = 2) [50, 52], with the remaining studies reporting all male participants (n = 168) ranging from the ages of 13 [55] to 36 years [42]. Five studies focused on the cricket skill of batting [42–45, 53], five focused on bowling [49, 50, 52, 54, 55] and one on the skill of fielding [51]. Seven studies involved adults of differing skill levels; four from an elite level (state cricket and above) [43, 51, 52, 54], and three from a sub-elite level (district cricket/semi-professional) [42, 44, 49]. Three studies involved juniors, two from elite environments [45, 53] and one from a school-based environment [55]. All studies involved the provision of augmented feedback within the intervention, with only one study exploring multiple types of augmented feedback [45]. Mechanisms of feedback provided included verbal [44, 53], visual [42, 43, 51, 52], or a combination [45, 49, 50, 54, 55].

## Methodological quality and risk of bias

Table 2 outlines the methodological quality assessment appraisal results from all of the studies included in the review. Initial assessment resulted in 97% agreement between assessors, with consensus achieved after discussion. Rankings for 'good', 'fair' and 'poor' methodological quality based on the Kennelly grading system was five, four and two respectively. Common limitations among the studies included the lack of recruitment of representative sampling, blinding of participants and randomisation of participants. Further, only one study [42] provided a power calculation to ensure that the study was adequately powered. One study was deemed to have high risk of bias when critical appraisal results were graded using the specified questions from the Downs and Black checklist [50]; however, the study was not included in any meta-analysis.

## Meta-analyses

The first meta-analysis (four studies) reported changes in anticipation accuracy (Fig 2) following an occlusion-based anticipation intervention that involved augmented feedback [42–45].

**Table 2. Critical appraisal of studies for quality assessment and risk of bias.**

| Reference | Critical appraisal score (out of 28) | Kennelly rating | Risk of Bias |
|---|---|---|---|
| Barker et al. [49] | 17 | Fair | Low |
| Brenton et al. [42] | 21 | Good | Low |
| Brenton et al. [43] | 20 | Good | Low |
| Fuss et al. [50] | 13 | Poor | High |
| Hopwood et al. [51] | 17 | Fair | Low |
| Middleton et al. [52] | 17 | Fair | Low |
| Muller et al. [44] | 20 | Good | Low |
| Neil et al. [53] | 14 | Poor | Low |
| Ranson et al. [54] | 20 | Good | Low |
| Smeeton et al. [45] | 20 | Good | Low |
| Wallis et al. [55] | 19 | Fair | Low |

Modified Downs and Black [35] checklist was used for the assessment of methodological quality and risk of bias. For quality assessment comparisons, study appraisal scores were graded as: good (≥20), fair (15–19), or poor (≤14) based on Kennelly et al. [39].

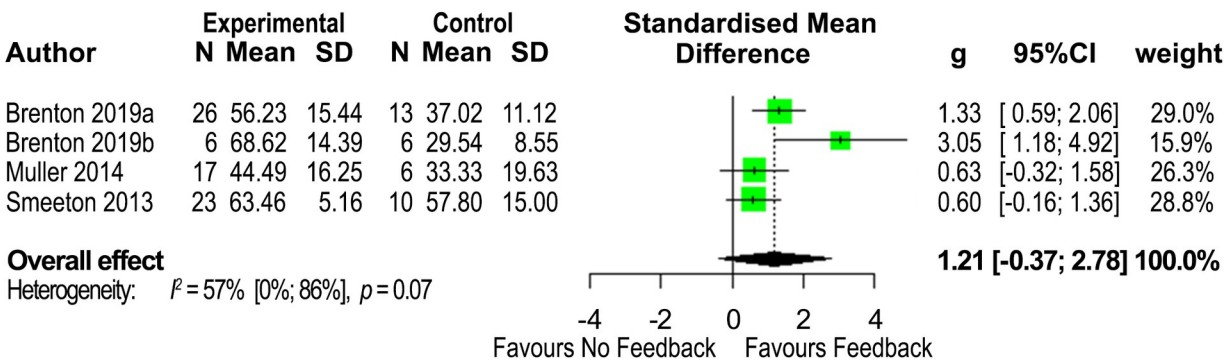

**Fig 2. Forrest plot displaying the *Hedge's g* effect sizes and 95% confidence intervals (CIs) for the perception attribute of anticipation, exploring accuracy among cricketers for feedback-involved intervention groups compared to no-feedback control groups.**

Although meeting the inclusion criteria, one study [51] was not included as additional data, after contacting the authors, was not available due to the length of time since conducting the study. Intervention group versus control group comparison showed an overall effect size (*Hedge's g*) of 1.21 (95% CIs [-0.37, 2.78]) in favour of the feedback-involved intervention group. Heterogeneity was deemed to be medium ($I^2$ = 57%, 95% CIs [0%, 86%], $p$ = 0.07) and no outliers were detected.

The second meta-analysis explored the action outcome of performance success (Fig 3), and included four studies [42–44, 51]. Intervention group versus control group comparison showed a medium overall effect size (*Hedge's g*) of 0.55 (95% CIs [-0.39, 1.50]) in favour of the feedback-involved intervention group. Heterogeneity was deemed to be low ($I^2$ = 14%, 95% CIs [0%, 87%], $p$ = 0.32) and no extreme effect sizes were detected.

## Discussion

This systematic review with meta-analysis had dual aims. The primary aim was to examine the impact of augmented feedback on cricket-related skill-based performance outcomes within existent literature. Eleven articles met the inclusion criteria, five of which were included in meta-analyses. Despite the systematic approach to searching, no studies with a standalone feedback-arm to their intervention were identified. Rather, the review focuses on feedback-involved interventions, and their impact on skill-based performance outcomes within cricket-specific research. All studies included in the meta-analyses [42–45, 51] explored anticipation

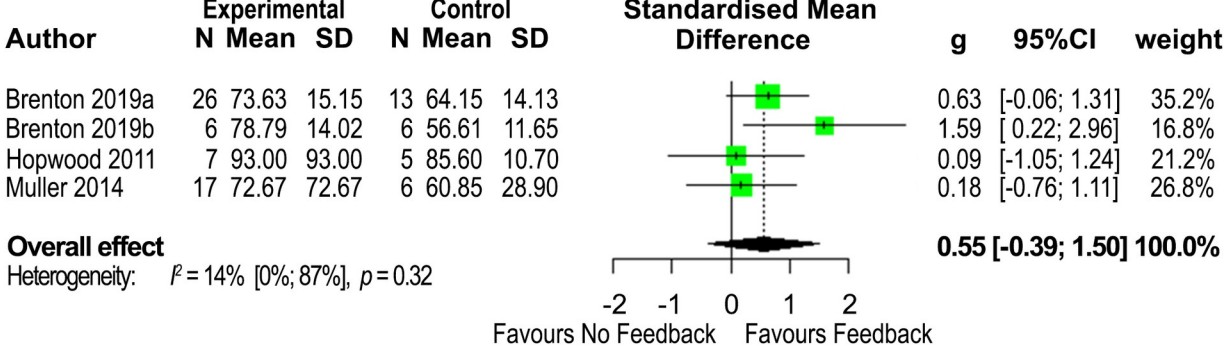

**Fig 3. Forrest plot displaying the *Hedge's g* effect sizes and 95% confidence intervals (CIs) for the action outcome measure of performance success, among cricketers for feedback-involved intervention groups compared to no-feedback control groups.**

ability employing temporal visual occlusion procedure, and consisting of an intervention that explicitly included augmented feedback to supplement learning. In most studies, augmented feedback was provided in the form of unoccluded footage following an occluded trial, either on its own [42, 43, 51] or alongside other forms of feedback [45]. Only one study utilised verbal feedback alone as a part of the training intervention [44]. The main findings of the meta-analyses, albeit not statistically significant, highlighted improved anticipation performance for the feedback-involved intervention groups for attributes of perception (anticipation accuracy) and action (performance success), compared to no-feedback control groups. Beyond the meta-analyses, this review sought to explore the included studies from an ecological dynamics perspective, with the aim of deriving recommendations and future research directions for using augmented feedback (with or without technology) for learning and teaching skill in cricket.

## Effect of augmented feedback on perception and action outcomes

The two meta-analyses explored the concept of perception-action coupling; a key process within the ecological dynamics theoretical framework for expertise acquisition [22]. All studies included in the meta-analyses investigated interventions featuring augmented feedback, to explore anticipation behaviour in cricket. Skilled anticipation is the ability to detect the information that specifies the forthcoming event or guides the unfolding action (for example, the ability to accurately detect the 'length of a delivery' prior to ball release from the bowlers run-up, arm, and hand motion) [56]. Anticipation is often studied using the temporal occlusion paradigm. This involves edited video footage or the use of occlusion-goggles, so that the visual information presented to the individual is more or less limited (for example, a bowler's run-up might be occluded 1-second before ball release). Superior anticipation ability is seen to be advantageous, especially in fast-ball sports such as cricket [57]. It is often assumed that the less viewing time needed to make a correct decision (such as if a ball will be delivered short or full) is used as an indication of anticipation skill [58].

In the existent research uncovered in this review, augmented feedback used the temporal occlusion paradigm with the aim of improving anticipation skill. Augmented feedback was provided using unoccluded footage following an occlusion trial [42, 43, 45, 51]. In one study [45], augmented feedback largely consisted of verbal knowledge of results, with two additional feedback intervention groups also receiving either video-based feedback of unoccluded footage, or further verbal instructions relating to imagery and cue utilisation. Only one study used goggle-based occlusion, with augmented feedback consisting of verbal information provided following occluded trials [44]. Studies explored a range of different levels of expertise, including elite (n = 6), sub-elite (n = 43) and youth-based (n = 23) cricketers. All studies included in the meta-analyses were deemed to be of good quality with low risk of bias (Table 2).

Results exploring anticipation accuracy (Fig 2), an attribute of perception ability, suggested a large effect size (*Hedge's g* = 1.21) in favour of the intervention group in studies exploring the skill of batting (n = 72 participants). However, the results are not statistically significant, likely affected by the small number of studies and the corresponding small sample sizes. Further, given the wide confidence intervals, caution should be used when interpreting results. Despite some heterogeneity ($I^2$ = 57%), no sub-group analysis was possible due to the limited number of studies, and no extreme effect sizes were identified as influencing between study heterogeneity. In considering why there was a lack of significance, another possibility may be that the feedback provided was not sufficiently guiding in drawing attention to key information sources.

For example, previous research has demonstrated positive effects of guided augmented feedback interventions using the occlusion paradigm on anticipation skill of soccer

goalkeepers [59]. In this study the gaze locations of expert goalkeepers, when viewing the run-up of a penalty-shot taker, were used to successfully guide and train beginners. An alternative explanation may also be that, at least in studies that used video displays [42, 43, 45, 51], the information available was somewhat impoverished. Exploring anticipation ability among skilled and less-skilled batters, Runswick et al. [57] demonstrated the importance of contextual information in addition to visual information. While the skilled batters showed greater overall anticipation ability, the use of contextual information was deemed more important prior to when early ball flight information became available. Given that key contextual information can often be present within the natural (sports) environment [56, 60], the capability to learn in these contexts within the above studies [42, 43, 45, 51] may have been supressed (see also, Mann et al. [61]).

Similar to anticipation accuracy, four studies explored the impact of the feedback-involved intervention groups on improvements to performance success (Fig 3) in batting tasks [42–44] and a fielding task [51]. Including elite (n = 13) and sub-elite (n = 43) cricketers, the results again favoured the intervention groups that received feedback, with a medium effect size (*Hedge's g* = 0.55); however, were not significant. Although all studies were of acceptable quality (good or fair) and with a low risk of bias, the results were again impacted by limited studies and small sample sizes. Another explanation for lack of significance worth considering may be the nature of the task completed.

In a previous meta-analysis, Travassos et al. [62] showed that the nature of the action used (called the 'requisite response') has a strong effect on the significance and magnitude of outcomes when comparing beginners and experts. Actions that required a full/representative response (e.g., intercepting the ball by hitting it) were associated with larger effect sizes compared to responses involving button pressing or verbalisations [62]. Only two studies performed skills in-situ [44, 51], requiring full representative responses to stimuli (such as a moving ball). However, within the Muller et al. [44] study, only the 'with-movement' group required a representative response; the 'without-movement' group only responding to the stimuli (a cricket delivery) through verbal indication of the appropriate movement. While operationally efficient, the laboratory context of two studies [42, 43] included in this meta-analysis may have further influenced the magnitude of the effect of the interventions used. Another problem (as discussed by van der Kamp et al. [56]) often encountered when using video-based studies, or 'without-movement' groups, is how perception is studied in isolation from action (i.e., the 'natural' information-movement coupling cannot be established). For instance, it has been widely demonstrated that requiring the individual to act changes the nature of information pick-up and the timing of actions that are performed [63, 64].

Despite large/medium effect sizes, the overall results should be interpreted with caution given the small number of studies identified, and the wide confidence intervals of the meta-analyses. Despite pooling all the available studies to explore augmented feedback in cricket, the results are still uncertain and therefore, support the need for further research to be completed. However, considering the possible methodological concerns raised (in particular, with respect to the nature of the tasks used that involved an augmented feedback intervention), the following section of the review considers augmented feedback within an ecological dynamics theoretical framework, with reference to how included studies have explored elements of this framework (summarised in Fig 4). To consider future research directions, the role of representative design in how augmented feedback can be utilised to help foster more appropriate and/or accurate coupling of information to movement is elaborated on, and finally, how technology may contribute to the delivery of feedback.

## Representative design: The foundation of creating functional augmented feedback opportunities

The athlete-environment relationship is a key tenet of the ecological dynamics framework [22, 67]. Acknowledging coupling between perception and action, a strength of the framework resides in underscoring the importance of the learners ability to adapt movement to the available information within their performance environment. To facilitate functional coupling [16, 68], coaches are tasked with designing representative practice settings. That is, affordance-rich practice landscapes with key sources of information made available [69], with relevant levels of variation [23]. Therefore, designing a representative learning practice environment enables coaches to appropriately embed augmented feedback opportunities, helping the learner explore the environment and develop functional information-movement couplings [69].

Only three studies [42–44] in this review incorporated elements of representative design. For example, despite utilising video-display of a bowler, both Brenton et al. [42] and Brenton et al. [43] required the batter to physically play the appropriate shot (i.e., move forward or backward) in assessing anticipation. Further, both studies enabled the 'with-movement' groups to re-bowl the cricket delivery they had experienced; providing another opportunity to directly explore the movement pattern and the information presented within the augmented feedback [42, 43]. Similarly, Müller et al. [44] used 'live' bowlers on an outdoor wicket when implementing an occlusion paradigm; allowing participants in the 'with-movement' group to face and play each delivery (while the 'without-movement' group only observed the 'live' delivery, and verbally responded with the appropriate movement response). The use of the natural (sports) environment enabled learners to consider feedback (presented on delivery type) with respect to information relevant to the competitive context. Such approaches have previously been suggested to improve both the representative nature of the task and subsequent exploration of perception-action coupling, which can lead to attunement to information sources more appropriate for the competitive context [70].

The majority of studies [45, 49–55] included in this review however, adopted approaches to intervention design that decouple (or establish an indirect relation between) perception and action processes (Fig 4). As discussed earlier, studying perception and action separately impacts subsequent movement patterns. For example, within cricket-related research, both Renshaw et al. [71] and Pinder et al. [72] demonstrated altered movement patterns for cricket batters facing a 'live' bowler compared to ball projection machines, where information from the bowler's run-up to couple the timing of the batter's action is absent (and only the ball trajectory information can be used). While altered movement patterns may not necessarily degrade overall performance, designs that lack key information sources and opportunities to act on them curtail the opportunities for augmented feedback to support exploration of task-appropriate information-movement couplings, which are presumably, only possible when key couplings unfold over time and are direct. In summary, representative learning design impacts the feedback opportunities that a coach or technology can present a learner and should be a requisite condition for its implementation.

## The role of augmented feedback and technology in supporting athlete-environment exploration

When provided appropriately, augmented feedback is an important coaching tool that supports the learner to develop skills, increase movement capabilities, and improve performance outcomes [6]. Ecological dynamics views augmented feedback therefore as an instructional constraint that influences the search processes of the perceptual-motor landscape [66, 73]. The role of augmented feedback (and technology more broadly) is seen as supporting the learner

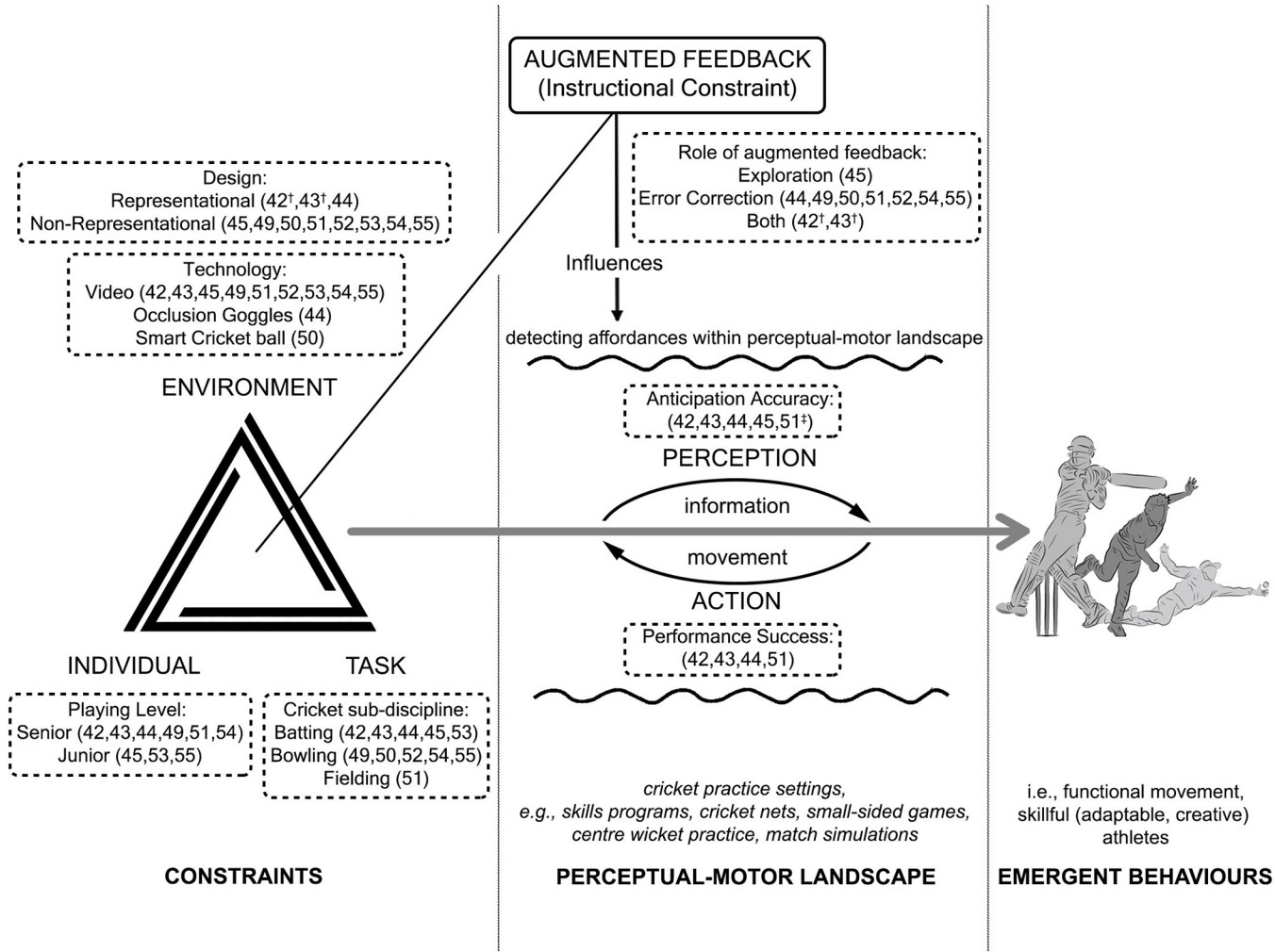

**Fig 4. Schematic depicting the ecological dynamics framework and the role of augmented feedback, with reference to studies included in the review.** Adapted from Davids et al. [65] and Newell et al. [66]. †some elements of; ‡not included in meta-analysis due to unavailability of data.

to progressively attend to their highly complex and constraints-driven performance environment [4, 24, 74]. According to Woods et al. [24], "technology use supports direct engagement and interaction between agents and their environments–promoting the development of experiential knowledge". This implies technology-based augmented feedback can promote different solution pathways, highlighting different sources of information previously unexplored or unrecognised by the learner [75]. This in turn can hasten the search process of the learner, as they are directed to explore more specific (as opposed to global) areas of their performance environment [17]. However, how technology is integrated into the feedback process needs to consider the principles of representative design and warrants further research.

The majority of included studies incorporated technology, often video-based, to facilitate the provision of augmented feedback [42, 43, 45, 49, 51–55]. However, only one study [45] utilised technology-led augmented feedback to facilitate the learner's exploration of their performance environment. Here, a guided discovery approach was used, in which the learner was provided instruction on what to attend to within their performance environment. Specifically, advanced cue information (on where the batter should focus in order to support recognition of ball spin direction) was supplemented with video-based replays or an imagery script.

Improved ability to explore the performance environment facilitates learning [17], but also gives rise to more adaptable learners and performers [69, 76]. For instance, athletes able to efficiently explore are better able to meet everchanging constraints (such as changing conditions of the pitch, game situation or opposition strategies in cricket [77]), and may also yield discovery of novel, creative solutions [78, 79].

A further study [50] explored the role of technology embedded within the performance environment. Utilising a smart cricket ball (technology-modified equipment), the single participant (a spin bowler) was profiled in-situ to attain performance and skill-based information (e.g., relating to spin rate, torque, and efficiency) often restricted to lab-based environments (via 3D motion capture). Following, the coach was able to provide augmented feedback during the training intervention based on this technology-derived information, aimed to improve these attributes and overall skill performance. Although unclear the specifics of how the technology was utilised during the training intervention, from an ecological dynamics perspective, technology such as this would be viewed as a *support opportunity* if harnessed appropriately [18]. Specifically, this type of smart technology could be used to explore bowling delivery adjustments to enable the athlete to self-regulate movement solutions. Such technology developments provide coaches with greater opportunity and flexibility when presenting feedback information to their athletes [80]. Given the continual advances to technology, including within sport settings, exciting opportunities exist for the development and use of technology which can be embedded within representative practice environments. Such technologies can support coaches to present augmented feedback in-situ; an important consideration which may help guide athlete exploration processes.

## Limitations and future directions

A limiting factor within this review was the small number of studies meeting the inclusion criteria. Therefore, the results of the meta-analysis may be impacted from sparse data bias [81]. However, previous research has indicated minimal studies still present a valid opportunity for performing quantitative analysis [82]. Further, *Hedge's g* was used to interpret effect sizes as it is more appropriate for use with smaller sample sizes [83]. Additionally, of the included studies, none explicitly explored the role of isolated augmented feedback. Rather, feedback-involved interventions and their subsequent impact on skill-based performance outcomes within cricket-specific research were explored. Such limitations in exploring key attributes of feedback (such as the timing, frequency and type) may have further impacted homogeneity of the studies and subsequent findings. Given the limited current research, the inclusion of the meta-analyses exploring occlusion-based anticipation attributes was justified due to the important role augmented feedback played within the training interventions to supplement learning. Overall, the limited augmented feedback research demonstrates a substantial gap in the literature within the sport of cricket. Future research should explore the explicit role that feedback has on improving skill development and performance, through exploring the impact of various attributes (e.g., *what*, *how*, and *when*). Further, the review highlighted the limited cricket cohort explored; identifying that future research should focus on both male and female cricketers to best understand the value of feedback practices. Finally, given the vast developments in technology, its use in the provision of feedback in novel, yet informative ways consistent with learning and teaching theory, should also be explored.

## Conclusion

To the authors knowledge, this is the first systematic review or meta-analysis that has explored the role of augmented feedback within research for the sport of cricket. The five studies

included in the meta-analyses explored the impact of an intervention involving augmented feedback on anticipation ability, indicating improvements to performance of perception (anticipation accuracy) and action (performance success) outcomes. However, the small number of participants in these meta-analyses, and wide confidence intervals, suggests a degree of caution should be used when interpreting the results. Given none of the included studies isolated the provision of augmented feedback, further research is required to fully elucidate our understanding of the role of augmented feedback in improving skill-based performance in cricket. Additionally, given the increasing availability and use of technology for presenting augmented feedback, exploring the role of technology enabled (augmented) feedback within an applied learning and teaching framework is warranted. We propose utilising an ecological dynamics framework to understand the potential positive aspects of technology and feedback on skill learning in cricket. As a starting point, we have highlighted the application of feedback technologies within a representative learning design context that promotes exploration of functional information-movement couplings. Therefore, the outcomes and perspective provided by this systematic review with meta-analysis provides an important contribution to understanding the impact of augmented feedback not only in the sport of cricket, but also the broader implications within the motor control theoretical framework of ecological dynamics.

## Supporting information

**S1 Checklist. PRISMA checklist.**
(DOCX)

**S1 Table. Database search string.**
(DOCX)

## Acknowledgments

The authors would like to thank Kathryn Duncan for her assistance and advice with the initial search strategy for this review.

## Author Contributions

**Conceptualization:** Kevin Tissera, Amanda C. Benson.

**Data curation:** Kevin Tissera.

**Formal analysis:** Kevin Tissera, Dominic Orth, Minh Huynh, Amanda C. Benson.

**Investigation:** Kevin Tissera, Dominic Orth, Minh Huynh, Amanda C. Benson.

**Methodology:** Kevin Tissera, Minh Huynh, Amanda C. Benson.

**Project administration:** Kevin Tissera, Amanda C. Benson.

**Supervision:** Dominic Orth, Amanda C. Benson.

**Visualization:** Kevin Tissera.

**Writing – original draft:** Kevin Tissera, Dominic Orth, Minh Huynh, Amanda C. Benson.

**Writing – review & editing:** Kevin Tissera, Dominic Orth, Minh Huynh, Amanda C. Benson.

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
