## [Decision Letter · Decision Letter 0]

14 Sep 2022

PONE-D-22-15875The impact of augmented feedback (and technology) on learning and teaching cricket skill: A systematic review with meta-analysisPLOS ONE

Dear Dr. Kevin Tissera,

Thank you for submitting your manuscript to PLOS ONE. After careful consideration, we feel that it has merit but does not fully meet PLOS ONE’s publication criteria as it currently stands. Therefore, we invite you to submit a revised version of the manuscript that addresses the points raised during the review process.

We would appreciate receiving your revised manuscript within 60 days. When you are ready to submit your revision, log on to https://pone.editorialmanager.com/ and select the 'Submissions Needing Revision' folder to locate your manuscript file.

To enhance the reproducibility of your results, we recommend that if applicable you deposit your laboratory protocols in protocols.io, where a protocol can be assigned its own identifier (DOI) such that it can be cited independently in the future. For instructions see: http://journals.plos.org/plosone/s/submission-guidelines#loc-laboratory-protocols

We look forward to receiving your revised manuscript.

Kind regards,

Chiara Milanese

Academic Editor

Journal Requirements:

“The lead author was supported by an Australian Government Research Training

Program (RTP) Scholarship. No other funding was received.”

5. We note that this manuscript is a systematic review or meta-analysis; our author guidelines therefore require that you use PRISMA guidance to help improve reporting quality of this type of study. Please upload copies of the completed PRISMA checklist as Supporting Information with a file name “PRISMA checklist”.

Reviewers' comments:

Reviewer's Responses to Questions

**Comments to the Author**

1. Is the manuscript technically sound, and do the data support the conclusions?

Reviewer #1: Yes

Reviewer #2: Yes

2. Has the statistical analysis been performed appropriately and rigorously? 

Reviewer #1: Yes

Reviewer #2: Yes

3. Have the authors made all data underlying the findings in their manuscript fully available?

Reviewer #1: Yes

Reviewer #2: Yes

4. Is the manuscript presented in an intelligible fashion and written in standard English?

Reviewer #1: Yes

Reviewer #2: Yes

5. Review Comments to the Author

Reviewer #1: Dear Authors

Conducting a systematic review on the impact of extended feedback (and technology) on learning and teaching cricket is a good idea in my opinion. The authors did a very interesting meta-analysis. Methodologically, the study is fine. The review is conducted in accordance with all the canons applicable to this type of research. In the conclusion it is clear what the methodological procedure is and what the results are.

Suggestions:

- I find PubMed missing from the databases listed. Of course, they are all selected correctly, and perhaps this would be duplicated, but it is missing for me.

- The section on conclusions is interestingly described. The authors pointed out the fact that the analysis performed was limited. I would suggest creating a separate section on conclusions and a separate section on limitations of the analysis.

I suggest accepting the paper for publication with minor corrections.

Reviewer #2: This study conducted a systematic review and meta analysis on the effectiveness of feedback-involved interventions on skill-based performance outcomes in cricket. The authors indicate that there is a lack of studies conducted directly on feedback and cricket skills, and instead examine research interventions which utilise augmented feedback in their methodological design. The discussion then uses an ecological dynamics lens to discuss the studies methodology and results. The following comments are made in the spirit of helping improve the quality of the manuscript.

General comments

There are a number of limitations which should be included in the manuscript, likely stemming from the lack of feedback specific studies in this area, and include the inability to review the frequency, timing and provision of feedback approaches on cricket skill. The review itself is rather an overview of feedback-involved interventions on skill-based performance outcomes in cricket-specific “research”.

It was disappointing to see that all studies in this space have only involved male cricketers. It is suggested that this point be highlighted, and include a call for future research in this space to include both male and female cricketers.

Specific Comments

Line 33-36: Suggest rephrase (For example, smart device applications that enable quick analysis

of movement patterns, such as the bat lift in cricket [3], providing augmented information that can be delivered as feedback (either directly to a player or by a coach), influencing how practice can translate into improved performance outcomes during competition)

Line 39: Please include: It “can” be delivered… after an action, (that is, it can also be provided during an action; also referenced in line 42 and again line 110

Could the authors explain why the timing of feedback (concurrent or terminal) was not delimited in table 1, unlike how the feedback type was included?

Could the lack of significance within the results be due to the fact that few included studies were designed to examine the effect of feedback on cricket performance?

Line 317 – The authors state “In considering why there was a lack of significance, another possibility may be that the feedback provided was not sufficiently guiding in drawing attention to key information sources.” Could this not also be due to only a few papers specifically examining the influence of feedback on performance?

Line 475 “Additionally, of the included studies, none explicitly explored the role of isolated augmented feedback.” Strongly suggested to include in limitations that the timing, frequency and method of providing feedback was not assessed in this study which may also have impacted homogeneity of the studies and subsequent findings.

Line 489: “the role of augmented feedback within the sport of cricket.” This review is just focused on research conducted within the sport of cricket

It is recommended the manuscript be reviewed generally to improve the quality of some sentence structures. (e.g., line 464: Given the continual advances to technology, including within sport settings, exciting opportunities exist for the development and use of technology which can be embedded within representative practice environments and, that supports coaches to present augmented feedback in-situ which helps guide athlete exploration processes)

6. PLOS authors have the option to publish the peer review history of their article (what does this mean?). If published, this will include your full peer review and any attached files.

Reviewer #1: No

Reviewer #2: No

---

## [Author Response · Author response to Decision Letter 0]

9 Oct 2022

Editor Comments 

1 Please ensure that your manuscript meets PLOS ONE's style requirements, including those for file naming 

The authors have checked the document and submission to ensure it meets the submission guidelines 

2 We note that the grant information you provided in the ‘Funding Information’ and ‘Financial Disclosure’ sections do not match. When you resubmit, please ensure that you provide the correct grant numbers for the awards you received for your study in the ‘Funding Information’ section 

Thank you for the comment. No grant was provided for this submission – the funding information referred to relates to a PhD stipend scholarship funded by the Australian Government that we are acknowledging. 

3 Please state what role the funders took in the study. If the funders had no role, please state: "The funders had no role in study design, data collection and analysis, decision to publish, or preparation of the manuscript." If this statement is not correct you must amend it as needed. Please include this amended Role of Funder statement in your cover letter; we will change the online submission form on your behalf 

The suggested comment will be included during the submission progress and cover letter to highlight that the funders played no role in the manuscript, given the nature of the funding (i.e., PhD stipend from the Australian Government) 

Comment to be added: 

4 In your Data Availability statement, you have not specified where the minimal data set underlying the results described in your manuscript can be found. PLOS defines a study's minimal data set as the underlying data used to reach the conclusions drawn in the manuscript and any additional data required to replicate the reported study findings in their entirety. All PLOS journals require that the minimal data set be made fully available. For more information about our data policy, please see http://journals.plos.org/plosone/s/data-availability. Upon re-submitting your revised manuscript, please upload your study’s minimal underlying data set as either Supporting Information files or to a stable, public repository and include the relevant URLs, DOIs, or accession numbers within your revised cover letter. For a list of acceptable repositories, please see http://journals.plos.org/plosone/s/data-availability#loc-recommended-repositories. Any potentially identifying patient information must be fully anonymized. We will update your Data Availability statement to reflect the information you provide in your cover letter 

All data used for the meta-analysis is provided in the manuscript. No additional data is available. 

5 We note that this manuscript is a systematic review or meta-analysis; our author guidelines therefore require that you use PRISMA guidance to help improve reporting quality of this type of study. Please upload copies of the completed PRISMA checklist as Supporting Information with a file name “PRISMA checklist” 

Thank you for the comment. A completed PRIMSA checklist has been included in the submission (Supplementary Table 1). The file name will be renamed as suggested 

Reviewer 1 

1 I find PubMed missing from the databases listed. Of course, they are all selected correctly, and perhaps this would be duplicated, but it is missing for me.

Thank you for the comment. PubMed was not selected as the authors had access to MEDLINE, which encapsulates the largest component of PubMed resources. Therefore, it is anticipated that articles that would have been located through PubMed would have been captured through the MEDLINE database search, in addition to other databases explored. We agree that it would be a duplication. 

2 The section on conclusions is interestingly described. The authors pointed out the fact that the analysis performed was limited. I would suggest creating a separate section on conclusions and a separate section on limitations of the analysis.

The review explores the limitations of the analysis in more detail (Lines 472-476), whilst briefly referring to it in the conclusion section (Lines 498-500) 

3 I suggest accepting the paper for publication with minor corrections 

Thank you for the positive comments on our manuscript. 

Reviewer 2 

1 There are a number of limitations which should be included in the manuscript, likely stemming from the lack of feedback specific studies in this area, and include the inability to review the frequency, timing and provision of feedback approaches on cricket skill. The review itself is rather an overview of feedback-involved interventions on skill-based performance outcomes in cricket-specific “research”. 

Thank you for the comment. The authors agree and have added to the manuscript to highlight these points relating to the nature of the review (feedback involved interventions) and limitations (feedback attributes) 

Nature of review 

– Lines 268-270 

“Rather, the review focuses on feedback-involved interventions, and their impact on skill-based performance outcomes within cricket-specific research.”

– Lines 477-479 

“Rather, feedback-involved interventions and subsequent impact on skill-based performance outcomes within cricket-specific research was explored.” 

Limitations of feedback attributes 

– Lines 479-481 

“Such limitations in exploring key attributes of feedback (such as the timing, frequency and type) may have further impacted homogeneity of the studies and subsequent findings.” 

2 It was disappointing to see that all studies in this space have only involved male cricketers. It is suggested that this point be highlighted, and include a call for future research in this space to include both male and female cricketers. 

This point has been added as a recommendation to guide future research 

Future directions recommendation 

– Lines 487-489 

“Further, the review highlighted the limited cricket cohort explored; identifying that future research should focus on both male and female cricketers to best understand the value of feedback practices.” 

Specific Comments

3 Line 33-36: Suggest rephrase (For example, smart device applications that enable quick analysis of movement patterns, such as the bat lift in cricket [3], providing augmented information that can be delivered as feedback (either directly to a player or by a coach), influencing how practice can translate into improved performance outcomes during competition)

Thank you for the comment. The sentence has been amended to have better flow 

Rephrase 

– Lines 33-37 

“For example, smart device applications that enable quick analysis of movement patterns, such as the bat lift in cricket [3], provide augmented information that can be delivered as feedback (either directly to a player or by a coach). This extrinsic information can subsequently influence how practice can translate into improved performance outcomes during competition [4].” 

4 Line 39: Please include: It “can” be delivered… after an action, (that is, it can also be provided during an action; also referenced in line 42 and again line 110

The sentence has been clarified 

Clarification 

– Lines 40-41 

“It can be delivered to a learner during or after an action is completed, putatively, to improve performance during later actions [4].” 

5 Could the authors explain why the timing of feedback (concurrent or terminal) was not delimited in table 1, unlike how the feedback type was included?

The timing of feedback is somewhat explored under the heading Feedback provided (i.e., it alludes to following a trial or video replay). The explicit nature of when feedback was provided was not explored given the limited scope of the studies included in the review (i.e., not standalone feedback studies which did not explore the timing as a key attribute of the feedback delivered). 

6 Could the lack of significance within the results be due to the fact that few included studies were designed to examine the effect of feedback on cricket performance?

The authors agree that the potential lack of significance may be due to the limited nature of how feedback was implemented and explored in the included research studies and would be something to investigate in further research. This limitation is included in the discussion section (Lines 318-320 and Lines 342-343) 

7 Line 317 – The authors state “In considering why there was a lack of significance, another possibility may be that the feedback provided was not sufficiently guiding in drawing attention to key information sources.” Could this not also be due to only a few papers specifically examining the influence of feedback on performance?

The author’s agree this could be the case. Please refer to our response to reviewer 2, point 6 above 

8 Line 475 “Additionally, of the included studies, none explicitly explored the role of isolated augmented feedback.” Strongly suggested to include in limitations that the timing, frequency and method of providing feedback was not assessed in this study which may also have impacted homogeneity of the studies and subsequent findings.

Thank you for the suggestion. The suggested comment has been included to identify these limitations within the discussion. 

Comment on limitation 

– Lines 479-481 

“Such limitations in exploring key attributes of feedback (such as the timing, frequency and type) may have further impacted homogeneity of the studies and subsequent findings.”

9 Line 489: “the role of augmented feedback within the sport of cricket.” This review is just focused on research conducted within the sport of cricket

The sentence has been amended to reflect this point. 

Clarification 

– Lines 494-495 

“To the authors knowledge, this is the first systematic review or meta-analysis that has explored the role of augmented feedback within research for the sport of cricket.” 

10 It is recommended the manuscript be reviewed generally to improve the quality of some sentence structures. (e.g., line 464: Given the continual advances to technology, including within sport settings, exciting opportunities exist for the development and use of technology which can be embedded within representative practice environments and, that supports coaches to present augmented feedback in-situ which helps guide athlete exploration processes) 

Thank you for the comment – the manuscript has been reviewed to assess sentence flow as advised. The highlighted sentence has also been refined as suggested. 

Rephrase 

– Lines 465-469 

“Given the continual advances to technology, including within sport settings, exciting opportunities exist for the development and use of technology which can be embedded within representative practice environments. Such technologies can support coaches to present augmented feedback in-situ; an important consideration which may help guide athlete exploration processes.”

---

## [Decision Letter · Decision Letter 1]

1 Dec 2022

The impact of augmented feedback (and technology) on learning and teaching cricket skill: A systematic review with meta-analysis

PONE-D-22-15875R1

Dear Dr. Kenin,

We’re pleased to inform you that your manuscript has been judged scientifically suitable for publication and will be formally accepted for publication once it meets all outstanding technical requirements.

Kind regards,

Chiara Milanese

Academic Editor

PLOS ONE

Additional Editor Comments (optional):

Reviewers' comments:

Reviewer's Responses to Questions

**Comments to the Author**

1. If the authors have adequately addressed your comments raised in a previous round of review and you feel that this manuscript is now acceptable for publication, you may indicate that here to bypass the “Comments to the Author” section, enter your conflict of interest statement in the “Confidential to Editor” section, and submit your "Accept" recommendation.

Reviewer #3: All comments have been addressed

Reviewer #4: All comments have been addressed

2. Is the manuscript technically sound, and do the data support the conclusions?

Reviewer #3: Yes

Reviewer #4: Yes

3. Has the statistical analysis been performed appropriately and rigorously? 

Reviewer #3: Yes

Reviewer #4: Yes

4. Have the authors made all data underlying the findings in their manuscript fully available?

Reviewer #3: Yes

Reviewer #4: Yes

5. Is the manuscript presented in an intelligible fashion and written in standard English?

Reviewer #3: Yes

Reviewer #4: Yes

6. Review Comments to the Author

Reviewer #3: It is an interesting and well written paper. The authors have adequately addressed the comments raised in a previous round of review. In my opinion, I feel that this manuscript is suitable for publication in PLOSONE in its current form.

Reviewer #4: (No Response)

7. PLOS authors have the option to publish the peer review history of their article (what does this mean?). If published, this will include your full peer review and any attached files.

Reviewer #3: No

Reviewer #4: No

---

## [Editor Report · Acceptance letter]

8 Dec 2022

PONE-D-22-15875R1 

The impact of augmented feedback (and technology) on learning and teaching cricket skill: A systematic review with meta-analysis 

Dear Dr. Tissera:

I'm pleased to inform you that your manuscript has been deemed suitable for publication in PLOS ONE. Congratulations! Your manuscript is now with our production department. 

Kind regards, 

on behalf of

Dr. Chiara Milanese 

Academic Editor

PLOS ONE